# Exploring Potential Ways to Reduce the Carbon Emission Gap in an Urban Metabolic System: A Network Perspective

**DOI:** 10.3390/ijerph19105793

**Published:** 2022-05-10

**Authors:** Linlin Xia, Jianfeng Wei, Ruwei Wang, Lei Chen, Yan Zhang, Zhifeng Yang

**Affiliations:** 1Key Laboratory for City Cluster Environmental Safety and Green Development of the Ministry of Education, Institute of Environmental and Ecological Engineering, Guangdong University of Technology, Guangzhou 510006, China; linlinxia@mail.bnu.edu.cn (L.X.); 2112024025@mail2.gdut.edu.cn (J.W.); zfyang@gdut.edu.cn (Z.Y.); 2Southern Marine Science and Engineering Guangdong Laboratory (Guangzhou), Guangzhou 511458, China; 3Guangdong Key Laboratory of Environmental Pollution and Health, School of Environment, Jinan University, Guangzhou 511443, China; 4State Key Joint Laboratory of Environmental Simulation and Pollution Control, School of Environment, Beijing Normal University, Xinjiekouwai Street No. 19, Beijing 100875, China; leichen@gdut.edu.cn (L.C.); zhangyanyxy@126.com (Y.Z.)

**Keywords:** network insight, urban carbon metabolism, center of gravity, carbon neutral paths

## Abstract

To meet the global need for carbon neutrality, we must first understand the role of urban carbon metabolism. In this study, we developed a land–energy–carbon framework to model the spatial and temporal variation of carbon flows in Beijing from 1990 to 2018. Based on the changes in carbon sequestration and energy consumption, we used ecological network analysis to identify the critical paths for achieving carbon neutrality during land-use changes, thereby revealing possible decarbonization pathways to achieve carbon neutrality. By using GIS software, changes in the center of gravity for carbon flows were visualized in each period, and future urban construction scenarios were explored based on land-use policy. We found that the direct carbon emission peaked in 2010, mostly due to a growing area of transportation and industrial land. Total integrated flows through the network decreased at an average annual rate of 3.8%, and the change from cultivated land to the socioeconomic sectors and the paths between each socioeconomic component accounted for 29.5 and 31.7% of the integrated flows during the study period. The socioeconomic sectors as key nodes in the network should focus both on their scale expansion and on using cleaner energy to reduce carbon emissions. The center of gravity gradually moved southward, indicating that the new emission centers should seek a greener mixture of land use. Reducing carbon emission will strongly relied on transforming Beijing’s energy consumption structure and increasing green areas to improve carbon sinks. Our results provide insights into carbon flow paths that must be modified by implementing land-use policies to reduce carbon emission and produce a more sustainable urban metabolism.

## 1. Introduction

Carbon emission reduction is an urgent issue to address the global climate change. The Paris Agreement proposed stabilization of the global temperature increase well below 1.5℃ above pre-industrial temperatures, thereby preventing extensive damage to the global ecosystem [1]. To meet the carbon neutrality challenge, countries are taking actions, but the proportion of energy consumption supplied by fossil fuels remains too high [2,3,4], the energy intensity (i.e., energy consumed per unit GDP) continues to increase [5] and the emission that results from land-use change grows year by year [6,7], especially in developing countries. This will create great challenges for these countries. Seeking effective ways to reduce carbon emission has therefore become a priority for governments around the world.

The rapid urbanization that is occurring in developing countries has promoted large-scale urban expansion, resulting in large negative environmental impacts for the areas that sustain these growing cities [8]. Particularly in China, unprecedented urbanization has greatly affected the urban system, and with urbanization expected to reach 70 to 75% by 2050 [9,10], the impact is likely to increase. In particular, enormous changes in the land-use and cover type (hereafter, “land use” for simplicity) have adversely affected the environment [11,12,13]. For example, these processes have been responsible for nearly 1.45 Pg carbon emission from 1990 to 2010 [14] and a decrease of about 101.79 TgC in carbon storage [15]. Currently, more than 70% of energy consumption still depends on fossil fuels in Chinese cities, and this makes decarbonization particularly challenging [3,16]. Vaughan (2020) noted that reducing carbon emissions and increasing green space to compensate for these emissions will be essential for China to achieve carbon neutrality while still maintaining rapid socioeconomic development [17].

Carbon neutrality is a widely accepted goal that can be achieved by following many paths, including afforestation, energy conservation, and emission reduction. Researchers have mostly focused on the aspects of an urban energy system to promote decarbonization of the energy structure and to reduce carbon emission sources [18,19]. Green compensation plays an important role to increase carbon sinks for offsetting carbon emissions during land-use management [20,21]. Researchers have also investigated the possibility of introducing carbon-neutral materials to achieve zero emissions during use of these materials [22,23]. Although technological innovation could provide significant ecological benefits, it is more cost-intensive than improving urban planning and has drawbacks such as long payback periods and facing innovative bottlenecks [24,25]. Thus, optimizing the land-use patterns offers an alternative and potentially more effective way to reduce carbon emissions for achieving carbon neutrality and improving sustainability [26].

Urban metabolism, which was first defined by Wolman (1965), provides tools for understanding the coordination and sustainable development of urban ecosystems [27]. Subsequent studies performed by Kennedy et al. (2007, 2010) concerned the carbon metabolic processes of different metropolitan areas and deciphered the impact of urban designing and transportation infrastructure on the urban metabolic processes in different cities [28,29]. Moreover, Kennedy et al. (2011, 2014) also pointed out the robustness of urban metabolism application to the low-carbon urban design [30,31]. In the context of growing contradiction between socioeconomic development and the ecological environment, urban metabolism studies have been extensively carried out for developing low-carbon urban ecosystems [12]. Many components of the urban metabolism, such as energy [32], water [33,34], land [35], carbon [36], nitrogen [37], and phosphorus [38], have been investigated. Perspectives such as the energy–carbon nexus [39,40,41,42] and land–carbon nexus [26] have been used to analyze the spatial characteristics and temporal changes of an urban carbon metabolism.

The linkage between urban metabolism and the land-use changes are essential to reveal the functional mechanisms of carbon transfers [43,44]. Substantial efforts have been devoted to characterizing the variabilities of urban carbon budgets associated with land-use changes, based on 3S technologies. For example, Hutyra et al. (2011) analyzed how spatial distributions of carbon sequestration and emissions were affected by urban expansion [45]. Moreover, a 4D-GIS database was used to trace material stock accompanied by urban development in cities of the UK and Japan [46]. These studies provided good examples of the interpretation of spatial urban metabolism. A recent study reviewed the state-of-the-art research on the spatialization of urban metabolism [47]. They show that recent spatial studies related urban carbon metabolism mainly focused on the modelling of spatial heterogeneity of utilities of urban resources (such as energy and land) and the resulting complex carbon flows or carbon footprint.

Due to the complex interactions among the components of an urban system and the frequency of the carbon flows [48,49], urban carbon metabolism based on land-use changes is seldomly analyzed regarding the changing center of gravity of carbon flows. Nevertheless, the gravity center could provide key information to recognize the spatiotemporal variation of carbon emissions related to land use [50,51,52], and also reflect the changing carbon sequestration of land cover in space, thus supporting carbon-neutral strategies [53]. Therefore, when the carbon flows related with land-use change can be quantified from a spatial perspective, feasible paths could be identified that offer a chance to modify a city’s urban carbon metabolism and thereby move towards carbon neutrality.

Ecological network analysis has become an effective tool that is widely used in urban metabolic analysis [41,54], from which the carbon exchanges within a metabolic network as a consequence of land-use changes can be recognized from a network perspective [55]. Examples include analyzing the spatial distribution and the transfer mechanisms for urban carbon metabolism; identifying the processes that increase or decrease carbon stocks [56]; and quantifying the spatial and temporal changes of carbon flows to understand an urban metabolic network. This perspective is important because different land uses have different carbon emission and sequestration characteristics, so changing the land use can greatly change the carbon balance. This approach also provides a macro-scale method to guide low-carbon development to balance a city’s carbon metabolism [57,58], thereby reducing or eliminating disorders in a city’s carbon metabolism.

In the current work, we set out to conduct a systematic study to visualize the migration of carbon flows related to land-use change and propose potential paths that cities can follow to achieve carbon neutrality. We also discuss how different land-use patterns moved the city either towards or away from carbon neutrality during the processes of urbanization. We combine the natural and socioeconomic components together into the urban metabolic system using the ENA method to quantify the variation of carbon flows, and to analyze the centers of gravity of carbon flows for future urban spatial planning.

The target Beijing was used as a case study owing to its representativeness. Beijing has undergone rapid urbanization and ranked as one of the most developed cities in China, accompanied by annually abundant energy consumption, large amounts of carbon emissions, and severe shortage of land resources. It is one of the few cities that has announced peak carbon emissions [59,60,61], in contrast to those megacities, such as Shanghai and Shenzhen, which are still facing increasing carbon emissions during the studied period [62,63]. Beijing city is experiencing urban shrink to increase carbon sinks, in contrast to most cities of China which are still expanding their urban body. Therefore, the study of the spatiotemporal variabilities of carbon metabolism taking Beijing as a case can provide critical information and knowledge for the peak carbon emissions of other cities. Using Beijing as a case study, we analyzed the urban carbon flows from 1990 to 2018 that were associated with land use and energy consumption changes. Moreover, we analyze changes in the center of gravity of carbon flows, which have been widely used as a good indicator to characterize the spatial variabilities of carbon flows during urban development, for estimating future urban spatial planning. This study is expected to support urban planning to reduce the gap between carbon emission and sequestration.

## 2. Methods

### 2.1. Model Framework

The models we used in this paper were developed in previous research by our research group [64]. Based on the availability of data, we defined eight nodes in the network: urban land, rural land, transportation and industrial land, cultivated land, forest, grassland, bodies of water, and bare land (i.e., unused land) including artificial, semi-natural, and natural types (Figure 1). We focus on the changes in direct carbon emissions caused by energy consumption, indirect carbon emission by the land use and natural carbon sequestration process for exploring carbon-neutral paths in the present study. Specifically, we set out to identify potential paths a city could follow to achieve carbon neutrality, and how different land-use patterns moved the city either towards or away from carbon neutrality during the process of urbanization. We identified the key nodes in the urban network during each period; analyzed the spatial and temporal changes of the integrated (direct plus indirect) and direct carbon flows; used GIS software to describe the spatial distribution of the carbon flows in each period; analyzed how the center of gravity for carbon flows moved over time; and discuss a scenario with different possible paths based on current urbanization plans.

We obtained data on land uses throughout Beijing from 1990 to 2018, at a 5-year interval for 1990–2015 and a 3-year interval for 2015–2018 at a spatial resolution of 30 m detailed in Section 2.5 and summarized the areas during each of the years in this period (Appendix A). We then added this information to version 10.5 of the ArcMap GIS software (www.esri.com (accessed on 22 March 2022)) to calculate the change in land use during each period. Based on the resulting land transfer matrices (Appendix A), we calculated the change in metabolic capacity of each land use (i.e., whether it was a net emission source or a net sequestration sink) to support ecological network analysis. Using these data, we identified land-use transitions that increased or decreased the city’s ability to achieve carbon neutrality. For instance, the path from transportation and industrial land to forest increased the land’s carbon sequestration potential and therefore helped to achieve carbon neutrality, whereas the reverse transition increased the land’s carbon emissions and therefore reduced the ability to achieve carbon neutrality.

We also used the GIS software to visualize the pattern of carbon flows to reveal the spatial variability of these flows and their changes over time. The analysis provided an explicit and integrated method for the natural activities of land with the energy consumption that occurs in that land to reveal the mechanisms that underlie an urban carbon metabolism.

### 2.2. Carbon Flow Calculations

The urbanization process causes land-use changes that increase or decrease carbon storage. The carbon calculations of direct carbon emission from transportation and industrial land, urban land, rural land and cultivated land were based on Chinese data on the consumption of each energy type detailed in Section 2.5, and the IPCC emission factors [65] are summarized in Appendix A. The accounting items were calculated as listed below:*C*_E*i*_ = *k*_E*i*_ × *m_i_*(1)
*C*_E*r*_ = *k*_E*r*_ × *p_i_*(2)
(3)$$C_{El}{=c}_{1}{f+c}_{2}{s+c}_{3}{a+cs}_{I}$$
where *C*_E*i*_ represents the carbon emissions from energy consumption, *C*_E*r*_, are the carbon emissions from respiration by the city’s population and livestock breeding and *C*_E*l*_ represents the carbon emissions from agriculture (carried by cultivated land); *k*_E*i*_ and *k*_E*r*_ represents the carbon emission coefficients; *m_i_* represents the consumption of energy type *i* (as standard tons coal equivalent, tce); *p_r_* represents the number of humans and livestock (Appendix A); *f* represents the mass of fertilizer; *a* represents the total use of agriculture machinery (represented by the engine power) ; *s* and *s_I_* represents the area of irrigated land and dry cultivated land; and the *c* values are the conversion coefficients (Appendix A). Then, carbon emission of transportation and industrial land can be calculated by Equation (1), and carbon emissions of urban and rural land could be calculated by the sum of *C*_E*i*_ and *C*_E*r*_.

The natural metabolic processes included carbon sequestration by natural activities, and were derived from four components (forest, grassland, water, and cultivated land). We used the following equation:*C_Si_ = k_i_ s_i_*(4)
where *C_S_* represents the carbon sequestration by natural components; *s* represents the area of the land use; and *k* represents the carbon sequestration coefficient of each natural component (Appendix A).

In the urban carbon metabolic network, we firstly calculated the carbon density of nodes:*w = C_i_/s_i_*(5)
where *w* represents the carbon density of nodes; for cultivated land, *C_i_ = C*_E*l*_
*− C_Si_*, for other land use types, and *C_i_* is the carbon metabolic rate of *C_E_* and *C_S_*.

Commonly, carbon flows are from *i* to *j* in an urban ecosystem and are shown as the flow (*f_ji_*) and the interactions among the network’s components are shown in Figure 1. These changes between land uses *i* and *j* can be described by the following equations:(6)$$f_{ji}=\Delta w_{ji}\Delta S_{ji}=\{\begin{array}{l} \;\left(w_{i}\;-\;w_{j}\right)\Delta S_{ji}\;(i\neq j)\;\\ \left(w_{i}+w_{j}\right)\Delta S_{ji}\;(i=j) \end{array}$$
where *f_ji_* (kg C year^−1^) represents the carbon flow from node *i* to node *j*; Δ*w_ji_* represents the differences in the carbon density between nodes along each path between nodes (i.e., different land-use transitions); Δ*S_ji_*, the area variation of the land use changing from *i* to *j*, is obtained from the land-use transfer matrices. Additionally, Δ*w_ji_* > 0 represents net carbon emissions; Δ*w_ji_* < 0 represents net carbon sequestration. To make outcomes more accurate, we applied standard deviation, mean value and 95% confidential intervals to evaluate the statistical differences. We calculated the mean value, standard deviation, and 95% confidence interval (CI) to account for the differences of carbon emission densities. Among these components, urban land fluctuated most obviously, ranging from 1.09 to 5.21 (95% CI) and deviating from its mean value by 47.1% (3.15 ± 1.48).

Subsequently, we also used ecological network analysis to calculate the integrated flows (direct plus indirect). The integrated input flows matrix **N** [66] was accounted for:**N** = (*n_ij_*) = **G**^0^ + **G**^1^ + ...**G***^k^ =* (**I** − **G**)^−1^
(7)
where *k* represents the number of components while 0 represents exchanges within a component; **I** represents the identity matrix; **G** represents the direct flow intensity matrix which is calculated by *g_ij_* = *f_ij_*/*T_j_*, where *T_j_* represents the throughflow of node *j*. Additionally, the dimensional integral flow matrix **Y** is calculated as **Y** = diag(**T**)**N** to represent the contribution weights of the sum of self-feedback and the direct effect, where **T** is the total integrated throughflow of each node. Detailed calculation procedures and the definition of carbon flows have been described previously [67].

### 2.3. Spatial Gradients for the Carbon Flows

We characterized the spatial changes of the urban carbon metabolism based on the cascade of land-use changes over time, which created gradients in the spatial distribution of the carbon metabolic processes. Many methods have been proposed to reflect the spatial heterogeneity of urban carbon metabolism, such as inversion of remote sensing data [68], spatial assignment based on accounting [67], and spatial interpolation [69] based on the sample plot observation. Considering the research objectives of the present study and feasibility, we selected the spatial assignment method based on carbon accounting to study the gradient changes for the integrated carbon flows. The different metabolic capacities of the different metabolic components in the carbon metabolic network changed the total direct flows and the integrated (direct plus indirect) input flows that created the spatial gradients. Based on the integrated flows, we considered the sums of the row vectors and column vectors for matrix **Y**, which reflected the weights of the contribution rates Σ*_i_ y_ij_* and Σ*_j_ y_ij_*, thereby obtaining the total direct and integrated input flows for each metabolic component. We then imported the results into the ArcMap software to calculate the spatial gradients and their distribution to reflect the changes in the total direct and integrated inputs from other metabolic components. Here, we defined the gradient as the change in carbon sequestration or emission between the start and end of a given period (e.g., high gradient category area representing rapid carbon transition). We used the Natural Breaks method provided by ArcGIS (www.esri.com (accessed on 22 March 2022)) to classify the optimal results automatically classified into five grades (Appendix A) based on the method of Xia et al. (2019a) [67].

### 2.4. Center of Gravity for the Carbon Flows

Through the migration of the gravity center of carbon flows in each gradient, we can explore the direction and location of carbon flows as a consequence of land-use changes. Therefore, to determine the spatial distribution and overall trend for the carbon flows, we used the center of gravity method [50]. We assumed that Beijing represented a homogeneous plane, so that the input flows for each metabolic component could be transformed into a point with a weight and so that the center of the region represented an equilibrium point based on the weights of the metabolic agents. The changes in the center of gravity over time indicated changes in the direct and integrated input flows throughout the study area, which reflected the spatial variation of the gradients that contributed to the direct and integrated input flows. We used ArcMap to create maps of the input flows based on the results of the direct and integrated flows. The location of the center of gravity was calculated as follows:*X_t_* = ∑*M_ti_ x_i_*/∑*M_ti_, Y_t_* = ∑*M_ti_ y_i_*/∑*M_ti_*(8)
where *X_t_* and *Y_t_* represent the longitude and latitude of the center of gravity in year *t*; *M_ti_* represents the input flows of gradient *i* in year *t*; and *x_i_* and *y_i_* are the longitude and latitude of the center of gravity for gradient *i* in year *t*.

### 2.5. Case Study and Data Sources

In this paper, we used Beijing as a case study to analyze changes in a city’s carbon metabolism network during urbanization. Beijing, China’s capital, has a large population (more than 7.5 × 10^6^ people throughout the study period), high energy consumption, and an extensive transportation network [70]. Apart from its unusually large size, it is a typical metropolis in China. From 1990 to 2018, Beijing’s total gross domestic product (GDP) increased by about 66 times, while the energy consumption increased by nearly three times [71]. Rapid land-use changes took place during the urbanization. Based on the land-use data in each year (Appendix A) and a digital elevation model (Appendix A), Beijing’s elevation was highest in the west and lowest in the east. The center of the urban land was mostly in the city’s southeastern plains region, and expanded towards the periphery of the southeastern plains. This is because this kind of terrain is highly suitable for urban development. The construction land developed rapidly at first, then its development slowed, leading to a large reduction in green areas (i.e., natural components of the city) and especially cultivated land [72], leading to a growing imbalance of the city’s carbon metabolism. After the city reached a mature stage, with relatively stable urban construction, the metabolism changed into a system with intensive resource utilization that produced different carbon flows in the process of intensification on account of the urgent needs to construct an ecological urban system. Thus, Beijing provides a good case study to illustrate which carbon flow paths caused by the rapid land-use changes would increase or decrease carbon emissions, and would therefore move the city farther from or closer to carbon neutrality. Finally, this analysis will reveal some possible paths the city could follow to help it towards carbon neutrality through appropriate changes in the city’s land-use policy.

The data used in our study included energy use, population, carbon emission coefficients, and the quantity of livestock for Beijing, which we obtained from the Beijing Statistical Yearbook [71,73,74], China Energy Statistical Yearbook [75,76,77,78,79] and IPCC (2006) [65]. The metabolic rates of the socioeconomic and natural components were calculated in kg C year^−1^. The spatially explicit land-use data were obtained at a scale of 1:100,000 for 1990, 1995, 2000, 2005, 2010, 2015, and 2018 from the Institute of Geographic Sciences and Natural Resources Research (https://www.resdc.cn/ (accessed on 22 March 2022)). We used ArcGIS to calculate a transition matrix between consecutive years to describe the change in land use and represent the flows in the city’s carbon metabolism. The source data for Figure 1 are freely obtained online (https://image.so.com/, (accessed on 22 March 2022)), and are from the above-mentioned databases for Figure 2, Figure 3, Figure 4, Figure 5, Figure 6 and Figure 7.

## 3. Results

### 3.1. Carbon Flows in the Urban Metabolic System

The total carbon emissions increased from 10.69 Mt C in 1990 to 20.61 Mt C in 2010, followed by steady decease after 2010 and finally reached 18.64 Mt C in 2018 (Figure 2a). Meanwhile, total carbon sinks varied insignificantly during 1990–2018, stabilizing at 0.38 Mt C year^−1^. At the peak in 2010, total carbon emissions were about 43.5 times higher than the total carbon sink, and this ratio remained high in 2018 (48.5 times). A serious imbalance between carbon emission and carbon sinks existed in Beijing throughout the study period. Meanwhile, the cumulative carbon emissions caused by the land-use and cover changes were 1.16 Mt C from 1990 to 2018 when the harmful flows of network (i.e., increased emission) contributed 15.37 Mt C year^−1^ and the beneficial flow (i.e., increased sequestration) contributed 14.21Mt C year^−1^ (Figure 2b). Beijing’s total direct and integrated carbon flows showed a volatile trend during the study period, with relatively high values of 5.81–8.07 Mt C year^−1^ from 1990 to 2000 and from 2010 to 2015, and with relatively low values of 1.08–1.97 Mt C year^−1^ from 2000 to 2010 and from 2015 to 2018. The changes in network flows (direct and integrated) were inconsistent with the changes in carbon emissions, depending on land sources consumed during urban expansion.

As the dominant components in Beijing’s carbon metabolism, transportation and industrial land contributed an average of 64.9% of the total carbon emissions, 46.0% of the total direct flows (beneficial and harmful flows), and 39.5% of the total integrated flows, respectively. As a result of the increasing energy consumption by industrial production and the transportation, storage, postal, and telecommunications services industries, emissions increased from 7.29 Mt C in 1990 to 11.06 Mt C in 2018. Due to the high metabolic density, slight changes in the area of transportation and industrial land caused large carbon flows, with an average contribution of 40.7%~51.3% to the direct network flows. During the study period, relative contributions of transportation and industrial land to carbon emissions decreased from 68.2% to 59.4% (Figure 2a). Meanwhile, its relative contributions to network flows decreased sharply from 51.8% to 9.1% (Figure 2b,c).

In urban land, the carbon emission experienced the largest increase from 2.01 Mt C in 1990 to 7.13 Mt C in 2018, mainly because of the industrial structure upgrading after 2005. In addition, ca. 2.2% of the total carbon emissions were caused by human respiration, which contributed 0.46 Mt C year^−1^ in 2018. Carbon emissions from rural land decreased sharply from 1.51 Mt C in 2015 to 0.38 Mt C in 2018, which mostly resulted from a large reduction in coal consumption which decreased carbon emission by 1.12 Mt C year^−1^. Large network flows produced by urban and rural land occurred from 1990 to 1995 and from 2010 to 2018 due to a reduction in the growth rate of urban land and a conversion of this land to transportation and industrial land. In addition, carbon emissions from cultivated land showed a continuous decreasing trend for the large reduction from fertilizer. The relative contributions of this land to network flows slightly increased as the land area sacrificed during urbanization.

For natural sectors, forest accounted for 91.3 ± 0.8% of the total carbon sink, and was the main reason for the peaking of carbon sink in 1995 (0.40 Mt C). In contrast, the carbon sink from grassland, bodies of water, and cultivated land only accounted for 3.1 ± 0.2, 4.9 ± 0.1, and 0.9 ± 0.2% of the total during the study period, respectively. Positive carbon flows were influenced by forest, grassland, and bodies of water, which contributed the positive carbon flows of 0.96, 0.51 and 0.28 Mt C year^−1^, respectively, during the study period. Total integrated flows from forest, grassland, and bodies of water accounted for averages of 7.3, 3.9, and 2.7%, respectively, of the total during the study period. Generally, integrated flows from these three components contributed relatively small flows to the network, but showed a large increasing trend, from 0.27, 0.14, and 0.08 Mt C year^−1^, respectively, from 1995 to 2000 to 0.77, 0.42, and 0.18 Mt C year^−1^ from 2010 to 2015. Most of the integrated flows for forest, grassland, and bodies of water were caused by integrated input flows from the socioeconomic sectors. As the urban expansion mainly derived from consumption of cultivated land, the natural components were slightly influenced by the socioeconomic sectors initially, but became more significantly influenced since 2015.

### 3.2. Spatial Distribution of Carbon Flows

To analyze how the carbon flows influence urban carbon metabolism and acquire a better understanding of the carbon metabolic network, we started by focusing on the direct flows. In Figure 3, the direct flows were generally dominated by the paths related to transportation and industrial land, which accounted for an average of 83.5% of the flows during the study period. The path transportation and industrial land → cultivated land, the first contributed path (i.e., the path with the largest flow), decreased from 1.17 Mt C year^−1^ from 1990 to 1995 to 0.01 Mt C year^−1^ from 2005 to 2010, then increased slightly to 0.47 Mt C year^−1^ from 2010 to 2015 and then decreased to 0.37 kt C year^−1^.

The major paths differed among the periods. From 1990 to 1995, direct flows were dominated by the urban land → transportation and industrial land path and the transportation and industrial land → cultivated land path, which accounted for 66.6% and 69.2% of the direct flows, respectively, from urban land and transportation and industrial land. The paths from transportation and industrial land to other land use types contributed 18.4% of the total direct flows. In contrast, the carbon flows paths from 1995 to 2000 were dominated by the rural land → transportation and industrial land path, the transportation and industrial land → cultivated land path, and the cultivated land → transportation and industrial land path, which contributed 87.9, 39.1, and 87.1% of the direct flows, respectively, from rural land, transportation and industrial land, and cultivated land. The direct flows from transportation and industrial land to other land use types increased to 26.1% of the total direct flows as the scale reduction existed in transportation and industrial land (i.e., causing massive carbon transition) which resulted in large carbon flows. This could explain the temporal changes of the harmful effects from transportation and industrial land during the study period. After 2000, the land use changes slowed, and the total direct flows were 0. 89 Mt C year^−1^ from 2000 to 2005 and 0.71 Mt C year^−1^ from 2005 to 2010. The paths in the two periods were dominated by the transportation and industrial land → cultivated land path and the cultivated land → transportation and industrial land path, which contributed 48.1 and 42.1% of the total direct flows, respectively. From 2010 to 2015, the total direct flows in the network increased to 3.55 Mt C year^−1^ and the main paths were dominated by the transportation and industrial land → rural land path and the cultivated land → transportation and industrial land path, which accounted for 42.7 and 65.4% of the direct flows, respectively, from transportation and industrial land and cultivated land. The paths from forest, grassland, urban land, and rural land to transportation and industrial land increased obviously compared to the flows from 2005 to 2010, which contributed 22.9% of the total direct flows. Interestingly, the changes of the direct flows between transportation and industrial land and forest were similar to the changes of the total direct flows, and contributed 11.1 and 13.4% of the total direct flows from 1995 to 2000 and from 2010 to 2015. From 2015 to 2018, the total direct flows decreased to 1.08 Mt C year^−1^, which were dominated by urban land → rural land and cultivated land → transportation and industrial land, which accounted for 36.2 and 31.1%, respectively, of the total direct flows.

Based on the analysis in Section 3.1, the total network flows decreased from 1990 to 2010, mainly due to the decrease in the integrated flows between the socioeconomic sectors and cultivated land and the decrease of the integrated flows between the socioeconomic and natural sectors, with decreases of 8.5 and 4.8%, respectively; these flows accounted for an average of 75.6 and 23.9% of the total integrated flows, respectively. On the other hand, the network’s total integrated flows decreased, mostly due to decreases in the integrated flows between the socioeconomic sectors and cultivated land, the integrated flows between the socioeconomic metabolic components, and the integrated flows between the socioeconomic and natural sectors, with decreases of 9.9, 7.5, and 10.8%, respectively, which accounted for an average of 39.8, 34.5, and 25.9% of the total integrated flows from 2010 to 2018. Figure 4 shows the variation in integrated flows in Beijing from 1990 to 2018. We found that the main path from 1990 to 1995 and from 2000 to 2005 was cultivated land → transportation and industrial land, which accounted for 35.2 and 42.5% of the total integrated flows, respectively. The paths from 1995 to 2000 and from 2010 to 2015 had similarly large carbon flows, dominated by the cultivated land → transportation and industrial land path (15.6%, 12.9%), the transportation and industrial land → cultivated land path (13.8%, 9.5%), the paths from transportation and industrial land to the natural sectors (11.5%, 9.4%), and the paths between the socioeconomic sectors (34.9%, 32.7%). The flows from 2005 to 2010 and from 2015 to 2018 were less than in the other periods, and were mainly influenced by the transportation and industrial land → cultivated land path, which accounted for 78.6 and 24.5% of the total integrated flows, respectively. In general, integrated flows through the network decreased at an average annual rate of 3.8%, mainly due to decreased carbon flows in the socioeconomic sectors. The main path from cultivated land to the socioeconomic sectors and the paths between each socioeconomic component accounted for 29.5 and 31.7% of the integrated flows during the study period.

The main paths in each period were mainly determined by Beijing’s land-use policies during the different developmental stages. The area of each land-use type changed from stable increases or decreases to fluctuation. Overall, the growth of carbon-neutral paths from 1990 to 2000 related to cultivated land and transportation and industrial land, mainly due to increases in the areas of transportation and industrial land. After China joined the World Trade Organization in 2001, the fast growth of economy further relied on the rapid urban expansion from 2000 to 2010. As the urban system gradually stabilized, the government began to prioritize low-carbon and green development, and managing the spatial distribution of these areas became priorities from 2010 to 2018. Thus, the increasing flows along the paths between socioeconomic components generally resulted from these more sustainable land-use policies. However, the flows from cultivated land to transportation and industrial land remained high after 2010 and led to high net carbon emissions.

### 3.3. Changes of the Centers of Gravity

#### 3.3.1. Changing Center of Gravity for Direct Flows

Based on the direct flow gradients in Appendix A, we calculated the spatial distribution of the gradient center of gravity of the direct input flows (Figure 5). From 1990 to 2018, the overall gradient center of gravity (i.e., the value for all gradient categories combined) for the direct input flows was located in the southeastern plains, up to 8 km from the north of the center (plains) and 8 to 14 km from the large areas of category I gradient, although some areas in this category existed north and west of the overall gradient center of gravity. Furthermore, the overall gradient center of gravity was at the intersection of other category gradients and the intersection mostly related to the areas with flows in the highest and second-highest categories (I and II), whereas the areas in category III were slightly scattered and far from the overall gradient center of gravity, followed by the second-lowest and lowest categories (IV and V, respectively). Therefore, the overall gradient center of gravity and its migration were mainly affected by categories I and II, followed by category III. During the study period, the location of the gradient center of gravity for the direct input flows moved southward by 14.4 km, towards the center of southeast plains, and reached the center of the southeast plains in 2010, after which it kept moving southward by about 1.35 km from 2010 to 2018.

The migration direction and distance for the overall gradient center of gravity changed over time. From 1990 to 2000, the center of gravity moved farthest (13.5 km) along an azimuth of 12°. The migration direction for the category I gradient (1400 to 5000 kt C year^−1^) moved about 7.5 km along an azimuth of 305°, but its migration distance was only about a half of that for the overall gradient center of gravity. The center of gravity for the category II gradient (760 to 1400 kt C year^−1^) moved along an azimuth of 55° and the category III gradient (235 to 760 kt C year^−1^) in the northeastern part of the mountainous area migrated along an azimuth of 330°. Thus, the overall trend was dominated by the combined effects of the category I, II, and III gradients. From 2000 to 2005, the overall gradient center of gravity continued to move northward, and its migration direction was relatively small. Compared to the previous period, the overall gradient center of gravity migrated along an azimuth of 200° over a distance of 6.8 km, which was mainly due to the decreased direct input flows in the southeastern plains. The category I gradient disappeared while the migration toward the overall gradient center of gravity weakened sharply, and the category II gradient therefore dominated the migration of the overall gradient center of gravity. From 2005 to 2015, the increased direct input flows in the center of the southeastern plains, causing the overall gradient center of gravity moving along an azimuth of 25° over a distance of 7.2 km, mainly as a result of the category I gradient’s center of gravity moving 4.5 km along an azimuth of 250° while the category II gradient’s center of gravity migrated along an azimuth of 165° over a distance of 8.1 km. From 2015 to 2018, the overall gradient center of gravity moved along an azimuth of 242° by 7.7 km, mainly as a result of migration of the category III gradient’s center of gravity along an azimuth of 120° over a distance of 35 km and migration of the category IV gradient’s center of gravity 18 km along an azimuth of 205°.

Generally, the occurrence of a category I gradient was mostly caused by the decreasing area of cultivated land and expansion of the area of transportation and industrial land from 1990 to 2000 and from 2005 to 2015. These both resulted in migration of the overall gradient center of gravity along an azimuth of 12° and 25°, mainly due to rapid land-use changes in the northern periphery of the southeastern plains. Meanwhile, the migration of the gravity center for the direct flow gradients would be mainly dominated by the changing land use in the northern part of Beijing, due to the contemporaneous expansion of land areas of Beijing Capital International Airport and restoration of water of bodies. However, the overall center of gravity moved along an azimuth of 205°, which suggests the need to improve the spatial distribution of carbon emission to promote low-carbon urban development; that is, it will be necessary to develop a subcenter to mitigate carbon emission from the center of the plains area. Although the migration of the center of gravity was related to the activities of the socioeconomic sectors, the urban system became increasingly mature. The land-use patterns gradually stabilized, which is why the category III and IV gradients influenced the overall gradient center of gravity in the period 2015–2018.

#### 3.3.2. Changing Center of Gravity for Integrated Flows

Based on the integrated input flow gradients shown in Appendix A, we mapped the spatial distribution of the overall gradient’s center of gravity (i.e., the value for all gradient categories combined) shown in Figure 6. From 1990 to 2018, the distribution of the input gradient center of gravity was similar to that of the direct input flows, which were also located in the north of the center of the southeast plains and the west areas of the northeast of the center (plains) with a category I gradient. As in the case of the direct input flows, the areas of intersection between gradients mostly occurred where category I, II, and II gradients occurred for the integrated input flows and the intersection with the gradient I center of gravity was largest, followed by the category II and III gradients. However, the distribution of the overall input gradient center of gravity were southeast of the direct input flows, which was closer to the category I and II input areas; that is, it was within 6 to 7 km north of the center of the southeastern plains, and another that was 4 to 12 km from the large areas of transportation and industrial land in the east. In addition, some category I and II areas existed in the northern and western parts of Beijing. The input gradient center of gravity generally moved southward; that is, it moved to areas closed to the center of the southeastern plains. The total movement was 9.1 km and the input gradient center of gravity shifted to an azimuth of 205°.

The migration distance and direction differed greatly between periods. The migration distance was greater and the migration direction different from 1990 to 2000 than from 2000 to 2010. The overall input gradient center of gravity migrated 9.4 km along an azimuth of 245°. This resulted from a large shift southwards of the category I gradient’s center of gravity, which was along an azimuth of 105° and 1.5 times the migration distance of the overall input gradient center of gravity but differed from the overall input gradient’s center of gravity direction, which was along an azimuth of 350° and migration of the category II gradient (13 km), which was close to that of the overall input gradient center of gravity with an azimuth of 10° but greater than the overall input migration distance (6.9 km) and the weakening impacts of the category III gradient, which moved 10.8 km in the opposite direction of the overall input gradient center of gravity. From 2000 to 2005, the overall input gradient center of gravity migrated southward by 4.4 m along an azimuth of 85°. This was mainly because the flow decrease in the southeastern plains was greater than that in the northwestern mountainous areas, and areas in the east with a category I gradient center of gravity moved 2.7 km along an azimuth of 355°. Due to the simultaneous decrease in the overall integrated flows in Beijing, a faster decrease occurred in the northwestern mountainous areas, which further weakened the migration of the gradient in northwest mountainous areas toward the overall input gradient center of gravity; that is, the centers of gravity of the category III gradient, category IV gradient (76 to 235 kt C year^−1^), and category V gradient (0 to 76 kt C year^−1^) all turned to the east. The nearly homogeneous distribution of gradients in the southeastern plains also caused the overall input gradient center of gravity to migrate east. From 2005 to 2010, the overall input gradient center of gravity continued to shift northeastward, by 7.2 km. The main reason for this migration trajectory was a further decrease of the integrated input flows in the southeastern plains caused by the disappearance of the category I gradient in the center of the study area and most of its periphery, and movement of the gradient II and III centers of gravity northward. From 2010 to 2015, the category I gradient was sufficiently common in the southeastern plains that migration of the category II gradient center of gravity could not offset the northward migration of the overall input gradient center of gravity, leading to a southwestward transfer. After 2015, the overall integrated input flows decreased obviously, leading to disappearance of the category I gradient in the southeastern plains. The overall input gradient center of gravity was mainly influenced by the category II and III gradients, which migrated southeastward.

## 4. Discussion and Policy Implication

### 4.1. Network Characteristics

Various groups have employed the ENA method to characterize the spatial and temporal changes of carbon flows from the network perspective on regional or city levels in China [26,57,58,64,80]. In the current work, we combined the ENA and GIS methods to estimate spatiotemporal changes in the gravity centers of carbon flows to bridge the relationship between urban form transition and land-use change-related carbon flows transfers.

Results indicated that the land-use structure has greatly changed the characteristics of Beijing’s carbon metabolic network. For example, the government’s policy of gradually moving heavy industries outside Beijing, which was implemented starting in 2000, should be modified to consider the possibility of optimizing the path from transportation and industrial land to cultivated land and other green types as other ways to approach carbon neutrality. Efforts to make these changes will depend on the spatial distribution of land use. In Beijing, this will be influenced by the “Master plan of Beijing: 2004–2020” and “Beijing 12th Five-Year Plan”, which have the goal of protecting cultivated land from the impacts of expansion of urban land and transportation and industrial land expansion, and by the “Master Plan of Beijing: 2016–2035”, which has the goal of increasing the city’s green areas to offset the increase of carbon emission. To these ends, the uncontrolled expansion of urban land, rural land, and transportation and industrial land will be replaced by more appropriate land uses for overall coordination, and this was the major reason that large negative carbon flows were produced by transportation and industrial land from 2010 to 2015 and by urban land from 2010 to 2018.

Despite the increase in the area of rural land from 2010 to 2018, urban intensification (to consolidate new residential construction with the goal of achieving a higher population per unit area) is an irresistible trend, which suggests that the increasing area of rural land would be an intermediate stage in the land use management in Beijing. Planners would be more likely to achieve a compact city to promote a reduction in urban and rural construction land, aiming to build a more ecologically and efficiently sustainable urban system [21,81]. Furthermore, we observed the trend of the compact city in the period of 2010–2015 and 2015–2018, which would have caused a reduction in urban lands by 19.3% (Appendix A). Therefore, as China has identified carbon neutrality as a crucial goal, highly developed cities such as Beijing will need to work harder to optimize the land-use patterns. Thus, urban planners should try to minimize uncontrolled urban expansion and instead focus on increasing the intensity of resource utilization (i.e., increasing its efficiency) to minimize carbon emissions.

### 4.2. Scenario Analysis

Analyzing the land use and emission trends provided insights into possible paths to bring Beijing’s metabolism closer to carbon neutrality. Based on the planning map in the “Master plan of Beijing: 2016–2035”, future land use maps (Figure 7) were created using GIS software. On the other hand, the energy indices used to create these maps were based on the “Prospect of Energy and Power Development in China (2020 Edition)”, in which non-fossil fuel energy was predicted to contribute 40 and 81% of the terminal energy consumption in 2035 and 2060, respectively (more detail are presented in Appendix A). On account of data from these scenarios from 2018 to 2035 and from 2035 to 2060, we predicted total direct carbon flows of 0.98 and 0.02 Mt C year^−1^, respectively, in Beijing’s carbon metabolic network. The corresponding total integrated flows were 1.12 and 0.02 Mt C year^−1^, respectively. From 2018 to 2035, the main paths were cultivated land → urban land (37.9%) and rural land → urban land (32.3%). From 2035 to 2060, the main paths were urban land → cultivated land (34.2%) and urban land → rural land (30.3%). Based on the areas of the different land use types in each year (Appendix A), the urban land evolved from fast expansion to a scale reduction, with the carbon metabolic intensity (Appendix A) increasing from 5.1 kg C/m^2^ in 2010 to 5.7 kg C/m^2^ in 2018 and then decreasing to 1.0 kg C/m^2^ in 2060. For the rural land, the carbon metabolic intensity sharply decreased from 1.4 kg C/m^2^ in 2010 to 0.2 kg C/m^2^ in 2018, mainly caused by eliminating coal consumption in rural land, and became roughly stable at this level from 2018 to 2060. Though the carbon emission from transportation and industrial land decreased, the direct carbon emission was still 2.10 Mt C in 2060, with emission of 8.3 kg C/m^2^ in 2060 (Figure 8).

Under these scenarios, the gradual maturation of the urban system is expected, leading to low-carbon urban development with increasing green compensation. From 2035 to 2060, forest is predicted to account for almost all of the total direct carbon flows with large proportion of integrated flows for offsetting the carbon emission. Although the energy transformation was conducted by decreasing the proportions of the 15 kinds of primary fossil fuel energy in Beijing, direct carbon emissions will decrease to 11.43 and 4.06 Mt C in 2035 and 2060, respectively, but carbon emissions will still be 26.8 and 9.1 times the carbon sinks in 2035 and 2060, respectively, and will represent carbon emissions 11.01 and 3.62 Mt C greater than the sequestration. Meanwhile, a recent study suggested that China could reduce its net carbon emission by 16% between now and 2060 by increasing carbon storage in forests [82,83]. For Beijing’s ecosystem, the total carbon sinks increased slightly during study period, mainly caused by the increasing forest area.

### 4.3. Policy Implications

After quantifying the carbon flows and their changes in response to changing land use, accurately visualizing the processes that underlie carbon transitions will help planners to explore the possible paths towards carbon neutrality. As our study shows, GIS technology permits characterization of the spatial patterns of carbon flows and permits analysis of the spatial pattern of the urban metabolism [84,85]. Our analysis revealed that the eastern parts of Beijing and the surrounding areas are important emission centers, and the southward migration of the overall gradient center of gravity indicated the development of a multi-center urban structure that has resulted in areas with a gradient of category I in the southeastern plains areas (Appendix A), which agrees with previous research [56,86]. By constantly moving carbon flows to the periphery of the built-up areas in the southeastern plains, the moving carbon emission to areas outside Beijing could relieve pressure on the city’s center. Currently, developing a multi-center urban structure has been shown to be an effective path to reduce carbon emission [87,88,89] and to improve the efficiency of energy use [90,91]. Beijing and other cities in China are facing the challenge of forming more compact urban systems to reduce pressure on their surrounding environment [60].

However, high energy consumption by the city’s socioeconomic sectors often leads to increased energy intensity [92,93,94]. As an important part of the urban planning [95], it will be necessary to learn how to avoid the effects of this increased compaction, which creates a risk of moving away from carbon neutrality. For example, planning more efficient transportation [15] strongly related to the key nodes of urban land and transportation and industrial land can improve the ability to develop a multi-center structure in Beijing. In addition, land use in the southeastern plains which links Beijing with the surrounding cities should be examined to determine the potential for strengthening the interaction with this region, since that may decrease the city’s carbon emissions [58], particularly if the transportation land at the periphery of the southeastern plains can be expanded to improve transportation efficiency. Thus, more attention should be paid to changing the development focus, which often involves the sacrifice of green spaces, thereby providing more sustainable urban planning [96]. For instance, transforming low-quality cultivated land with a low economic value and a low carbon-sink capacity to other land use types can be considerable to improve Beijing’s development in some cases.

Finally, although China’s commitment to tackle climate change is strong, it is still not clear how China can achieve its carbon-neutral vision [97,98]. Plans elsewhere in the world use a deadline of 2050, which is earlier than the 2060 deadline that has been set for China. Achieving carbon neutrality will rely on each city’s efforts to follow low-carbon paths, using strategies such as changing the current energy consumption structure [99,100,101] and developing more effective urban planning [102,103], improving the proportion of clean energy [100]. These paths are popular solutions in some regions to achieve carbon neutrality, and will also be vital to achieve carbon neutrality in China’s cities [97].

## 5. Conclusions

In this study, we performed a spatially explicit analysis of Beijing’s carbon flows from 1990 to 2018, as well as China’s proposed scenarios for 2035 and 2060, to explore the possible paths to reduce carbon emission reduction. The direct and integrated emission and sequestration by the city’s socioeconomic sectors and cultivated land in Beijing’s urban carbon metabolism were critical to achieve carbon neutrality, especially by regulating land transfers from cultivated land to socioeconomic sectors. The overall center of gravity migrated southward as a result of rapid land use change, leading to the development of a multi-center structure; the migration was driven primarily by migration of the centers of gravity for the category I and II gradients during the study period, thus providing clear information to help decision makers achieve carbon neutrality. Our analysis of the two future scenarios for Beijing revealed that the efforts of city planners to transform the city’s energy-consumption structure and increase green compensation will bring Beijing closer to carbon neutrality by reducing the gap between carbon emission and carbon sequestration. Therefore, future land use planning should consider how to increase the city’s green areas and improve their spatial distribution while constructing a more efficient urban system. However, it will be necessary to clarify the driving mechanisms that underlie the city’s urban carbon metabolism and to perform a sensitivity analysis of the network to determine the reliability of the network model for use in future predictions.

This study provides a network perspective to explore the paths to narrow the gap between carbon emissions and sequestrations. However, there are still some limitations. First, it is necessary to clarify the driving mechanisms that underlie the city’s urban carbon metabolism and to perform a sensitivity analysis of the network to determine the reliability of the network model for estimation in future predictions. Secondly, we rearranged the sub-sectors into each land use type owing to lack of relevant information. Therefore, further research should establish more detailed land-use data. Thirdly, a lack of land transaction data between cities limits fine land management. Usually, the local bureau of statistics will not directly report these data online, and for many cities, the data used for metabolic analysis on time series are scarce. The future carbon metabolism database should be able to promote modeling the urban metabolic system in the time cascades, and the future spatial distribution between urban metabolism and land-use policy should also be projected based on the research basis of various models.

## Figures and Tables

**Figure 1 ijerph-19-05793-f001:**
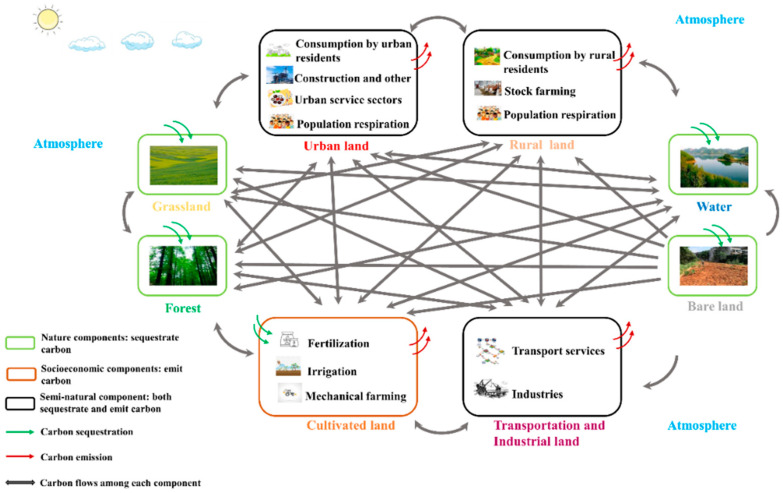
Framework for the urban carbon metabolic network used to study the effects of land-use change on the potential to achieve carbon neutrality. Note: the sub-processes in the red and black boxes show various socioeconomic activities causing carbon emissions. Carbon density of each socioeconomic land area was estimated by the sum of emissions from all the sub-processes dividing its corresponding land area. The red and green arrows represent the carbon exchange between atmosphere and biosphere. The black arrow represents carbon transitions between two components.

**Figure 2 ijerph-19-05793-f002:**
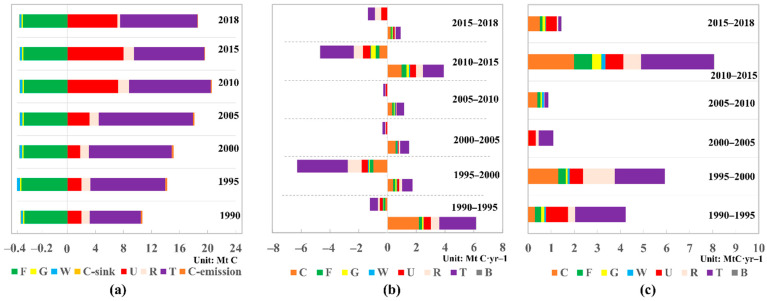
(**a**) Carbon emission and sequestration in each year. Direct (**b**) and integrated (**c**) carbon flows through the network in each period. Note: F, forest; G, grassland; W, bodies of water; U, urban land; R, rural land; T, transportation and industrial land; B, bare land; C, cultivated land; C-emission, carbon emission from cultivated land; C-sink, carbon sequestration produced by cultivated land.

**Figure 3 ijerph-19-05793-f003:**
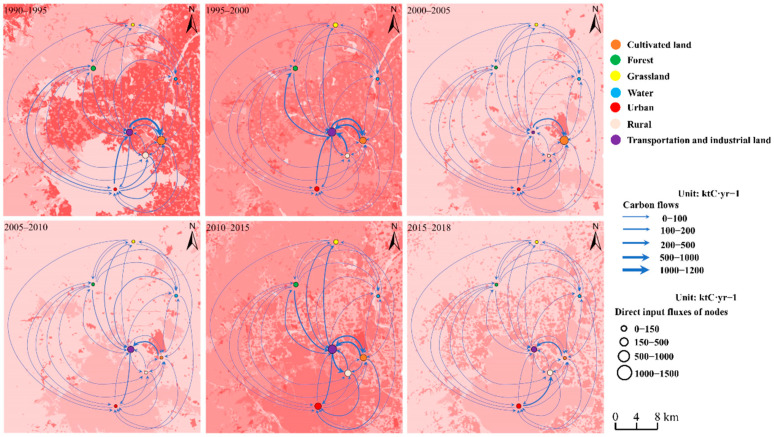
Spatial distribution of direct carbon flows in each period (the flows that are shown account for at least 80% of the total flows). Note: the flows related to bare land were too small to show in the figure.

**Figure 4 ijerph-19-05793-f004:**
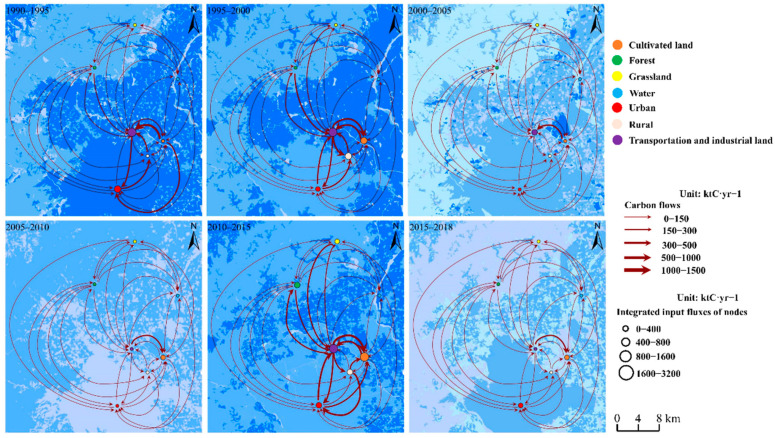
Spatial distribution of integrated carbon flows during each period (the flows that are shown account for at least 80% of the total flows). Note: the flows related to bare land were too small to show in the figure.

**Figure 5 ijerph-19-05793-f005:**
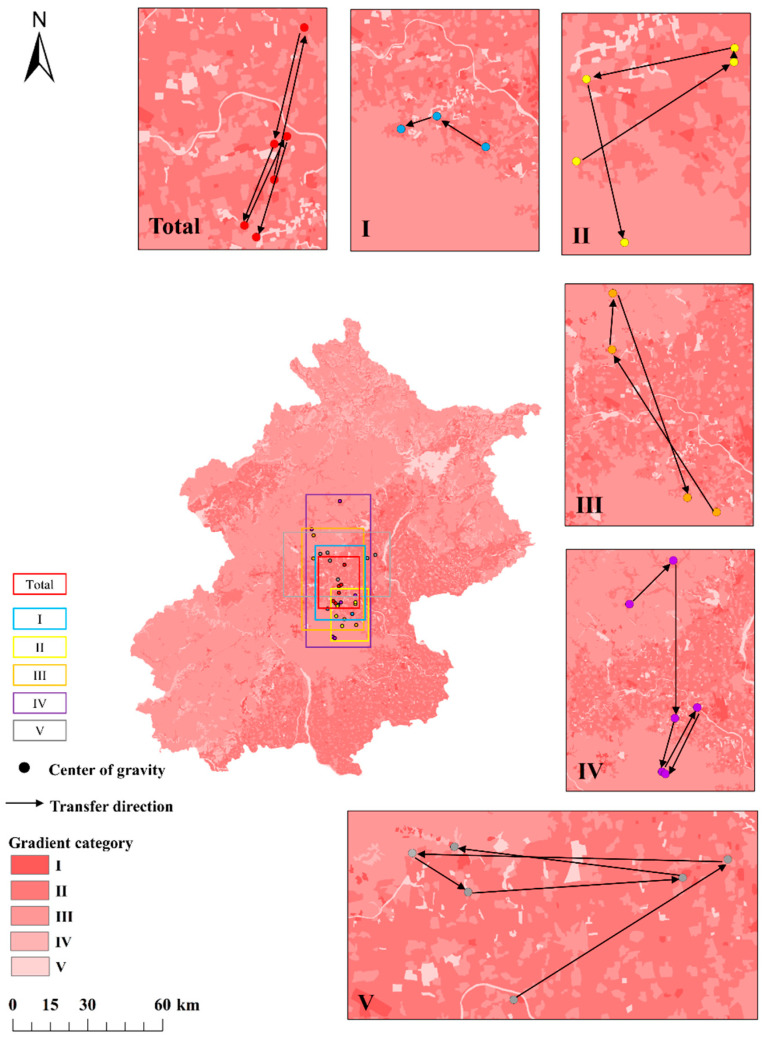
Migration of the center of gravity for the direct flow gradients. Note: gradient category I disappeared from 2000 to 2010 and from 2015 to 2018, gradient category II disappeared from 2015 to 2018, and gradient category III disappeared from 2005 to 2010; numbered rectangles refer to the gradient maps surrounding the overall map.

**Figure 6 ijerph-19-05793-f006:**
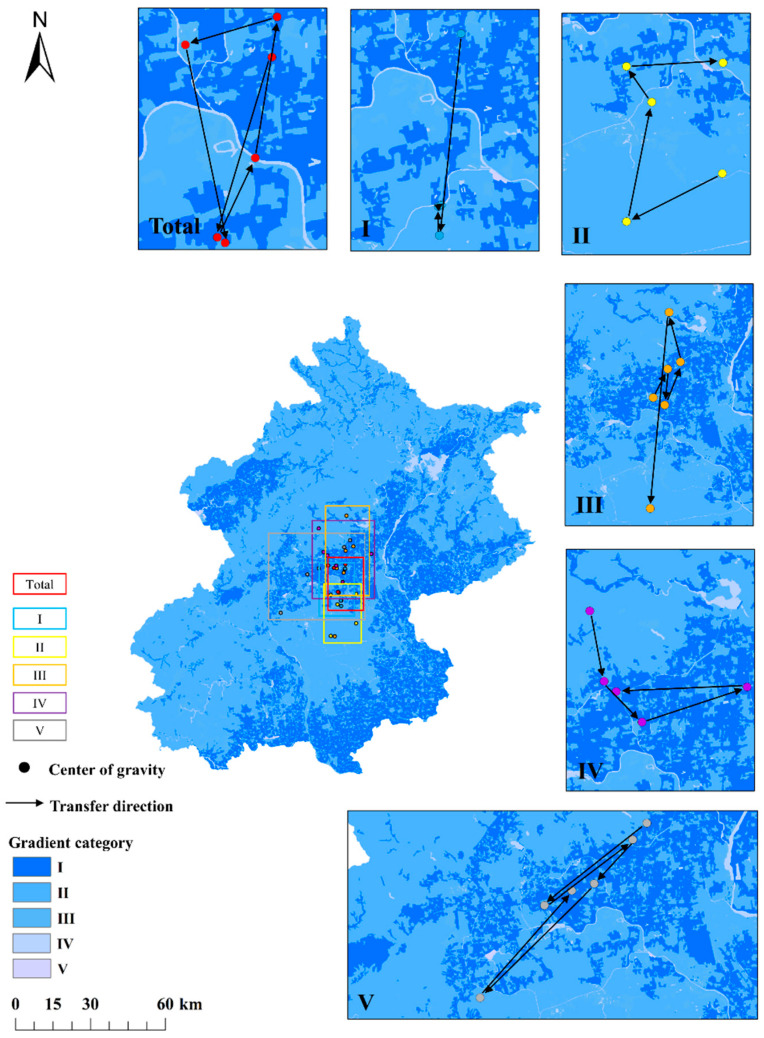
Changes in the location of the integrated gradient’s center of gravity. Note: Areas with gradient I disappeared from 2005 to 2010 and from 2015 to 2018; areas with gradients II or III disappeared from 1995 to 2000; Numbered rectangles refer to the gradient maps surrounding the overall map.

**Figure 7 ijerph-19-05793-f007:**
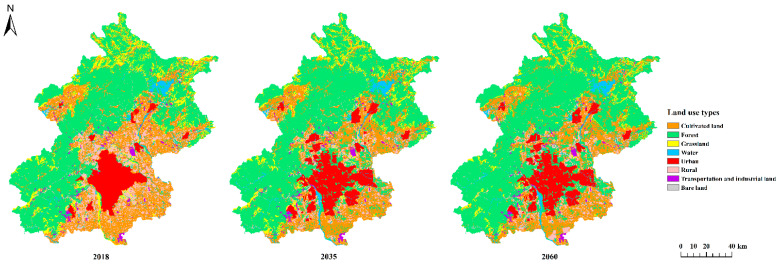
Land use maps in Beijing from 2018 to the two planned scenarios (in 2035 and 2060).

**Figure 8 ijerph-19-05793-f008:**
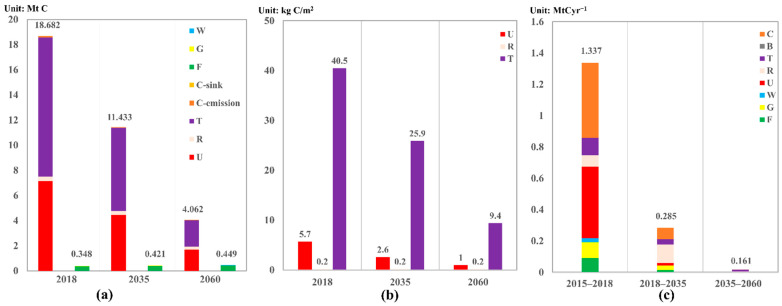
(**a**) Direct carbon emission and sinks in 2018, and predicted values in the 2035 and 2060 scenarios; (**b**) carbon metabolic intensity of the urban socioeconomic components in each year; (**c**) annual average integrated (direct plus indirect) flows from 2015 to 2018, from 2018 to 2035, and from 2035 to 2060. Note: F, forest; G, grassland; W, bodies of water; U, urban land; R, rural land; T, transportation and industrial land; B, bare land; C, cultivated land; C-emission, carbon emission from cultivated land; C-sink, carbon sinks produced by cultivated land.

## Data Availability

Not applicable.

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
