# Peer review of "Exploring Potential Ways to Reduce the Carbon Emission Gap in an Urban Metabolic System: A Network Perspective"

_ijerph, 2022, doi:10.3390/ijerph19105793_

Round 1

Reviewer 1 Report

The manuscript presents a clear overall aim, with a comprehensive description of the specific objectives, the requirements and how to address within the study. The specific objectives are  clearly set out and the overall feasibility of the work is convincingly addressed. The objectives are above the state of the art. The scope of the article  makes sure that the overall and the specific objectives are achievable.  The strategy to advance beyond from the state of the art well explained. The innovation aspects are beyond the state of the art that is properly evaluated in the paper to frame the expected results. The methodology issound and well formulated  More detailed explanation of migration of the center of gravity for the direct flows gradients shoul be added in the manuscript.

Author Response

Response: Thank you for your recognition and suggestions! We have added related explanation in lines 265-266. In addition, we further analyze the factors that cause the changes in direct flow gradients and discussed it in lines 516-519.

         We are grateful for your comments, and hope that our responses and the resulting changes in the manuscript will be satisfactory. However, we will be happy to work with you to resolve any remaining issues.

Reviewer 2 Report

Dear authors, I read your paper carefully. This study proposes and tests a land-energy-carbon framework to understand the spatial and temporal variation of carbon flows in Beijing from 1990 to 2018.  It focuses on urban planning rather than technological fixes as the way forward, but my main concern is that the way how it is written, the scale on which data is collected and GIS is applied, it is actually not so relevant for urban planners. 

For example, I am not sure why it is interesting to know the movement of the gravity center, at least not from the perspective of an urban planner. The way how you present the movement of the gravity center is something very abstract. Please explain how this abstract concept can be explained in real practical actions. Why do you not give names of the districts, places? Why do you keep it so abstract, with coordinates?

I am also confused why you would calculate a gravity center... while you also all for a multi-center structure?

Secondly, I am not impressed by your engagement with the literature. You mention urban metabolism, but you have not really engaged deeply with that. In the discussion, you do not refer back and do not explain how your research contributes to the wider body of urban metabolism research. You also do not explain how representative your study is, and how replicable. 

You also do not acknowledge the limitations of your model. Please write a small subsection on limitations and why it is (not) replicable. You state that Bejing makes a good case study... so why would other cities (in China, East-Asia, globally) do not make a good case study? Or do they?

Small comments

Please define "bare land". I am not sure what you mean with that. 

L36. "the most pressing issue", is it? What about biodiversity loss, water management issues? 

L75. The sentence is not complete. Or grammatical error. (There are more examples of this. Please do some language revision)

L89-L99. The information on urban metabolism is quite limited. Important names as Kennedy are not mentioned, or the acknowledgement of different approaches (e.g. industrial ecology, urban political ecology) are missing. Land use changes, urban planning... are also political. So why do you not include a political ecology perspective too?

L89-L99. The novelty of your paper is not so clear. You might add one sentence that spatially explicit methods were not that common to show spatial and temporal evolutions, until researchers like Tanikawa (e.g. 2009) introduced 4D-GIS.  

Also interesting: according to Bahers et al. 2022, the spatial turn is more recent.

Bahers, J.B., Athanassiadis, A., Perrotti, D. and Kampelmann, S., 2022. The place of space in urban metabolism research: Towards a spatial turn? A review and future agenda. Landscape and Urban Planning221, p.104376. 

However, they focused more on materials, while your research is building on the focus of flows and stock of carbon, which is an element. 

L82 I miss a definition of carbon flows. I know urban metabolism and can envision how materials "travel" in a city, but how should I envision an element traveling in a city?  

L115 What do you understand as "healthier" urban development?

L!128. Again, it is not so clear to me what is the novelty of your research... After "based on this review of the literature", you should state what is missing and how your research contributes to that. You write this in L141 etc.

L136. Although I have some idea, perhaps it is good to explain what you mean with gravity, or introduce with one sentence why this is important to know.

 L130-L140: This belongs in the method section, or could just be removed. L139-L140 could be transformed in a statement of your specific objective.

L157 So your data is from 1990, 1995, 2000, 2005, 2010, 2015, and 2018. What are the limitations and problems of having data about land uses for only every five years? Especially as a city like Beijing is changing so rapidly in the past decade... 

Figure 1. It is quite a complex figure. I suggest adding some extra information to the legend. Please explain to me why you would visualise a sink and source with an arrow. They are not flows...  

L567. What is "(in)appropriate"? 

L576. Please revise this sentence. 

L580 What exactly in your results lead to this conclusion that minimizing uncontrolled urban expansion might be a way forward? Or is this also the result of qualitative case study research? 

L603. "environment-friendly distribution of urban land uses" is an awkward expression. What do you mean with that exactly?

L604. Your sentence is not complete... 

Author Response

Review #2:

  1. It focuses on urban planning rather than technological fixes as the way forward, but my main concern is that the way how it is written, the scale on which data is collected and GIS is applied, it is actually not so relevant for urban planners. For example, I am not sure why it is interesting to know the movement of the gravity center, at least not from the perspective of an urban planner. The way how you present the movement of the gravity center is something very abstract. Please explain how this abstract concept can be explained in real practical actions. Why do you not give names of the districts, places? Why do you keep it so abstract, with coordinates?

Response: The key point of this work is to construct a framework to explore the maximum reduction potential of carbon emissions associated with land use changes, which can be achieved by adjusting the size, direction, and location of land use changes. These findings, combined with urban development policies, are used to investigate the application of the aggregated framework for urban planning. Indeed, the results of this work is not directly relevant for urban planners, it is instead to provide fundamental methodology support for urban expansion and shrink from the perspective of urban carbon metabolism.

       In line with the above explanation, we focused on the direction and location of land use change related carbon flow transfers, by studying the movement of the gravity centers. Similar researches have been conducted in previous studies such as Chen et al., 2021; Li et al., 2020; Meng et al., 2021. It has been concluded that the movement of the gravity centers is a good indicator to characterize the spatial variabilities of carbon flows during urban development. Moreover, we added the physical geological information of the main causes that are responsible for the changes of gravity centers, which make the results more visible. Examples are shown in lines 516-519.

       References:

Chen, L., Xu, L., Cai, Y., Yang, Z., 2021. Spatiotemporal patterns of industrial carbon emissions at the city level. Resour. Conserv. Recycl. 169, 105499.

Li, X., Wang, J., Zhang, M., Ouyang, J., Shi, W., 2020. Regional differences in carbon emission of China’s industries and its decomposition effects. J. Clean Prod. 270, 122528.

Meng, G., Guo, Z., Li, J., 2021. The dynamic linkage among urbanisation, industrialisation and carbon emissions in China: Insights from spatiotemporal effect. Sci. Total Environ. 760, 144042.

  1. I am also confused why you would calculate a gravity center... while you also all for a multi-center structure?

Response: The calculation of gravity center is to study the change of gravity center migration, and show how the change of carbon flow gravity center caused by land use changes can be accommodate with the land use policy. In fact, we calculated gravity centers for all the individual carbon flows. We find that the gravity centers of each level of carbon flows varied substantially during the urban development, towards a multi-center urban structure. In summary, our estimated results point to a multi-center carbon flows associated with the switch of single- to multi-centers of the city. This finding in turn proves the solidness of our estimation in lines 669-696.

  1. Secondly, I am not impressed by your engagement with the literature. You mention urban metabolism, but you have not really engaged deeply with that. In the discussion, you do not refer back and do not explain how your research contributes to the wider body of urban metabolism research. You also do not explain how representative your study is, and how replicable. 

Response: We have reorganized the Introduction section to make it easier to understand. The recent literature researches related to the field of urban carbon metabolism were reviewed and updated in lines 74-102. We also added contents related to urban metabolism to explore the further application of the framework into other cities in lines 586-591. In addition, we conducted sensitive analysis to analyze the replicability of the method in lines 226-231.

  1. You also do not acknowledge the limitations of your model. Please write a small subsection on limitations and why it is (not) replicable.

Response: Thank you for your suggestion! We have added a paragraph to describe the limitations of the model in lines 726-738.

  1. You state that Bejing makes a good case study... so why would other cities (in China, East-Asia, globally) do not make a good case study? Or do they?

Response: The target Beijing was used as a case study owing to its representativeness in the following aspects: 1) it undergone a rapid pattern of urbanization and ranked as one of the most developed city in China, accompanied by annually abundant energy consumption, large amount of carbon emissions, and severe shortage of land resource; 2) it is one of the few cities that has announced the peak carbon emission, in contrast to those megacities, such as Shanghai, Shenzhen, which are still facing the increase of carbon emissions during the studied period; 3) it is experiencing the urban shrink to increase carbon sink in contrast to most cities of China which are still expand in their urban body. Therefore, study of the spatiotemporal variabilities of carbon metabolism taking Beijing as a case can provide critical information and knowledge for the peak carbon emission of other cities.

We have added the above mentioned discussion in lines 132–144 and in lines 281-302.

Small comments

  1. Please define "bare land". I am not sure what you mean with that. 

Response: we have added the definition of "bare land" as the unused land which could be once used as constructed land and abandoned and thus without carbon emission and sequestration in a certain period in line 154.

  1. L36. "the most pressing issue", is it? What about biodiversity loss, water management issues? 

Response: We changed the sentence to “Carbon emission reduction is an urgent issue to mitigate climate change around the world.” in line 36.

  1. L75. The sentence is not complete. Or grammatical error. (There are more examples of this. Please do some language revision)

Response: We changed the sentence to “Green compensation plays an important role to increase carbon sink for offsetting carbon emissions during the land use management” in lines 64-65.

  1. L89-L99. The information on urban metabolism is quite limited. Important names as Kennedy are not mentioned, or the acknowledgement of different approaches (e.g. industrial ecology, urban political ecology) are missing. Land use changes, urban planning... are also political. So why do you not include a political ecology perspective too?

Response: We have updated recent literatures that reported on the urban metabolism, and summarized the results of these studies in lines 75-109. The added references are shown below:

Kennedy, C., Cuddihy, J., Engel-Yan, J., 2007. The Changing Metabolism of Cities. J. Ind. Ecology 11(2), 43-59.

Kennedy, C., Pincetl, S., Bunje, P., 2011. The study of urban metabolism and its applications to urban planning and design. Environ. Pollut. 159(8), 1965-1973. 10.1016/j.envpol.2010.10.022.

Kennedy, C., Steinberger, J., Gasson, B., Hansen, Y., Hillman, T., Havránek, M., Pataki, D., Phdungsilp, A., Ramaswami, A., Mendez, G.V., 2010. Methodology for inventorying greenhouse gas emissions from global cities. Energy Policy 38(9), 4828-4837. 10.1016/j.enpol.2009.08.050.

Kennedy, C., Stewart, I.D., Ibrahim, N., Facchini, A., Mele, R., 2014. Developing a multi-layered indicator set for urban metabolism studies in megacities. Ecol. Indicat. 47, 7-15. 10.1016/j.ecolind.2014.07.039.

Li, X., Wang, J., Zhang, M., Ouyang, J., Shi, W., 2020b. Regional differences in carbon emission of China’s industries and its decomposition effects. J. Clean Prod. 270, 122528.

Zhang, G., Zhang, N., Liao, W., 2018a. How do population and land urbanization affect CO2 emissions under gravity center change? A spatial econometric analysis. J. Clean Prod. 202, 510-523.

Moreover, the current work mostly pay attention to the interactions between urban form and urban carbon metabolism, which is a key concern issue raised by e.g., IPCC. Investigation of this issue requires multi-disciplines input, such as urban ecology, landscape ecology, and political ecology. We analyze the policy impact on carbon metabolism from the political ecology perspective in section 4. Discussion and implications in lines 627-641.

  1. L89-L99. The novelty of your paper is not so clear. You might add one sentence that spatially explicit methods were not that common to show spatial and temporal evolutions, until researchers like Tanikawa (e.g. 2009) introduced 4D-GIS.  Also interesting: according to Bahers et al. 2022, the spatial turn is more recent.

Bahers, J.B., Athanassiadis, A., Perrotti, D. and Kampelmann, S., 2022. The place of space in urban metabolism research: Towards a spatial turn? A review and future agenda. Landscape and Urban Planning221, p.104376. However, they focused more on materials, while your research is building on the focus of flows and stock of carbon, which is an element. 

Response: We have reorganized this paragraph to highlight the novelty of this work in lines 93-99, and the suggested references have been added in manuscript.

The added references are shown below:

Bahers, J.-B., Athanassiadis, A., Perrotti, D., Kampelmann, S., 2022. The place of space in urban metabolism research: Towards a spatial turn? A review and future agenda. Landsc. Urban Plan. 221, 104376. 10.1016/j.landurbplan.2022.104376.

Hutyra, L.R., Yoon, B., Hepinstall-Cymerman, J., Alberti, M., 2011. Carbon consequences of land cover change and ex-pansion of urban lands: A case study in the Seattle metropolitan region. Landsc. Urban Plan. 103(1), 83-93. 10.1016/j.landurbplan.2011.06.004.

Tanikawa, H., Hashimoto, S., 2009. Urban stock over time: spatial material stock analysis using 4d-GIS. Build. Res. In-format. 37(5-6), 483-502. 10.1080/09613210903169394.

  1. L82 I miss a definition of carbon flows. I know urban metabolism and can envision how materials "travel" in a city, but how should I envision an element traveling in a city?  

Response: We have added the description for the construction of special network of urban carbon metabolism in lines 168–172, and defined the concept of carbon flows in lines 219-221. In addition, the related references have been added.

  1. L115 What do you understand as "healthier" urban development?

Response: We intended to express a low-carbon urban ecosystem. The “healthier” herein is indeed inaccurate, and has been revised as “low-carbon urban ecosystem” line 82.

  1. L128. Again, it is not so clear to me what is the novelty of your research... After "based on this review of the literature", you should state what is missing and how your research contributes to that. You write this in L141 etc.

Response: This paragraph has been revised to make the novelty clearer in lines 125-148, as shown below:

In the current work, we set out to conduct a systematic study to visualize the migration of carbon flows related to land use change and propose potential paths that cities can follow to achieve carbon neutrality. We also discuss how different land use patterns moved the city either towards or away from carbon neutrality during the processes of urbanization. We combine the natural and socioeconomic components together into the urban metabolic system using the ENA method to quantify the variation of carbon flows, and to analyze the centers of gravity of carbon flows for future urban spatial planning.

The target Beijing was used as a case study owing to its representativeness. Beijing has undergone a rapid urbanization and ranked as one of the most developed cities in China, accompanied by annually abundant energy consumption, large amount of carbon emissions, and severe shortage of land resource. As one of the few cities that has announced the peak carbon emission (Liu et al., 2020a), in contrast to those megacities, such as Shanghai, Shenzhen, which are still facing the increase of carbon emissions during the studied period (Liu et al., 2020b; Zhou et al., 2018). Beijing city is experiencing the urban shrink to increase carbon sink in contrast to most cities of China which are still expand in their urban body. Therefore, study of the spatiotemporal variabilities of carbon metabolism taking Beijing as a case can provide critical information and knowledge for the peak carbon emission of other cities. Using Beijing as a case study, we analyzed the urban car-bon flows from 1990 to 2018 that were associated with land use and energy consumption changes. Moreover, we analyze changes in the center of gravity of carbon flows, which has been widely used as a good indicator to characterize the spatial variabilities of carbon flows during urban development, for estimating future urban spatial planning. This study is expected to support urban planning to reduce the gap between carbon emission and sequestration.

  1. L136. Although I have some idea, perhaps it is good to explain what you mean with gravity, or introduce with one sentence why this is important to know.

Response: This sentence has been revised as “Moreover, we analyze changes in the center of gravity of carbon flows, which has been widely used as a good indicator to characterize the spatial variabilities of carbon flows during urban development, for estimating future urban spatial planning.” in lines 144–146.

  1. L130-L140: This belongs in the method section, or could just be removed. L139-L140 could be transformed in a statement of your specific objective.

Response: This sentence has been revised as suggested in lines157–165.

  1. L157 So your data is from 1990, 1995, 2000, 2005, 2010, 2015, and 2018. What are the limitations and problems of having data about land uses for only every five years? Especially as a city like Beijing is changing so rapidly in the past decade... 

Response: We used the every five years data due to the following two aspects: 1) to accommodate with the “Five-Year Plan for National Economic and Social Development” issued by Chinese government. In addition, it is a common study way about the land use changes using every five years data by China’s research academy; 2) the impact of land use changes on the socioeconomic development and associated carbon emissions usually has a time lag. Therefore, it is suitable to analyze the temporal changes of carbon flows related to land use change in a certain time interval (every 5 year)

  1. Figure 1. It is quite a complex figure. I suggest adding some extra information to the legend. Please explain to me why you would visualise a sink and source with an arrow. They are not flows...  

Response: Figure 1 has been revised as suggested and added with additional information to the legend in lines 168-172. In addition, we added interpretation of each component and arrow and its color. In addition, we changed the “carbon sink” and “carbon source” to “carbon sequestration” and “carbon emission”, respectively, to characterize the carbon transfers between atmosphere and biosphere.

  1. L567. What is "(in)appropriate"? 

Response: We intended to express that the inappropriate area of the artificial land is mostly caused by the uncontrolled expansion that is harmful to the urban carbon metabolic system with large positive carbon flows (i.e., increasing the carbon emission). The appropriate land uses mean the reasonable urban spatial distribution under the policy regulation. We have revised “inappropriate” to “uncontrolled” in line 602.

  1. L576. Please revise this sentence. 

Response: We have revised the sentence in lines 609-610.

  1. L580 What exactly in your results lead to this conclusion that minimizing uncontrolled urban expansion might be a way forward? Or is this also the result of qualitative case study research? 

Response: This conclusion was based on the modelling results of the carbon flow transfers, which are influenced by the carbon metabolic density of each land component and the transferred area during land use change. We also conducted scenario analysis to predict the future spatial patterns of carbon metabolism. At last, we reached to the conclusion that minimizing uncontrolled urban expansion may be a way forward in lines 607-619.

  1. L603. "environment-friendly distribution of urban land uses" is an awkward expression. What do you mean with that exactly?

Response: We have revised the "environment-friendly distribution of urban land uses" to “the low-carbon urban development with increasing green compensation” in line 643.

  1. L604. Your sentence is not complete... 

Response: The sentence has been revised as “From 2035 to 2060, forest is predicted to account for almost all of the total direct carbon flows with large proportion of integrated flows for offsetting the carbon emission.” in lines 643-645.

We are grateful for your comments, and hope that our responses and the resulting changes in the manuscript will be satisfactory. However, we will be happy to work with you to resolve any remaining issues.

Reviewer 3 Report

This study assessed the impacts of land-use changes on carbon emissions in Beijing using data from 1990 to 2018. The authors estimated carbon flows over a network with multiple nodes corresponding to land-use types. The authors compared the spatial distributions of carbon flows for different periods and discussed the relationships between socioeconomic changes and carbon emissions. The impact assessment method used in this study is applicable to other cities and is useful for developing urban climate policies. Although the Methods section has some unclear points, the manuscript is basically well-written, and the calculation result is interesting. The Reviewer's comments are listed below.

Comment #1 (page 5, lines 183-184): According to Equation (2), the carbon emissions calculated by the authors include carbon emissions from the respiration of people living in Beijing. This calculation is unique. Most studies have paid little attention to physiological carbon emissions from human bodies because climate policies cannot reduce such carbon emissions. Did human respiration have a significant impact on the total carbon emissions in Beijing?

Comment #2 (pages 4-5): The usage of mathematical symbols in Section 2.2 is misleading. Please avoid using the same symbol to represent different meanings. The following points need to be corrected.

  • The index i of Equation (1) indicates energy type but does not in Equation (2). What does i of pi mean? The index i of Equation (5) indicates node.
  • w of Equation (5) should be wi because carbon intensity depends on node.
  • Roman and italic symbols are mixed; w of Equation (5) and w of Equation (6).
  • Lowercase and uppercase symbols are mixed; pi of Equation (2) and Pi on line 186.
  • The Reviewer could not understand the meaning of Equation (6). Please write the definition of ΔSji as a mathematical equation. How did the authors calculate ΔSji from si and sj?

Comment #3 (pages 15-17): The authors predicted carbon emissions in 2035 and 2060 based on a land-use scenario. In addition to land-use changes, population and economic growths are also strongly associated with carbon emissions. Please describe demographic and economic assumptions used in the carbon emissions prediction.

Author Response

Review #3:

  1. Comment #1 (page 5, lines 183-184): According to Equation (2), the carbon emissions calculated by the authors include carbon emissions from the respiration of people living in Beijing. This calculation is unique. Most studies have paid little attention to physiological carbon emissions from human bodies because climate policies cannot reduce such carbon emissions. Did human respiration have a significant impact on the total carbon emissions in Beijing?

Response: Thank you for your suggestion! The whole-process analysis of urban carbon metabolism has yet been addressed because urban metabolic systems are hybrids of natural and artificial structures. Optimal regulation of an urban metabolic system requires better definition of the nodes of the system, the paths between them, and the flows along those paths. Although human respiration contributed low amount of the total carbon emissions in 2018 (ca. 2.2%), it is a process that constitute the network of urban carbon metabolism in lines 352-353. Carbon emissions impact on the urban carbon metabolism is beyond our scope in the current study, whereas this part of carbon emissions should be received attention in future study of carbon neutrality.

  1. Comment #2 (pages 4-5): The usage of mathematical symbols in Section 2.2 is misleading. Please avoid using the same symbol to represent different meanings. The following points need to be corrected.
  • The index i of Equation (1) indicates energy type but does not in Equation (2). What does i of pi mean? The index i of Equation (5) indicates node.
  • w of Equation (5) should be wi because carbon intensity depends on node.
  • Roman and italic symbols are mixed; w of Equation (5) and w of Equation (6).
  • Lowercase and uppercase symbols are mixed; pi of Equation (2) and Pi on line 186.
  • The Reviewer could not understand the meaning of Equation (6). Please write the definition of ΔSji as a mathematical equation. How did the authors calculate ΔSji from si and sj?

Response: We have defined the meanings of the indices and corrected the symbols error in line 203. Also, we have clarified that the land-use map was generated from Landsat TM data in lines 309-311, and the ΔSji was mainly obtained from the land use transfer matrices through the change detection statistic of Arcmap GIS software, including the information of the area changes between node i and node j. We have added this information in lines 224-225.

  1. Comment #3 (pages 15-17): The authors predicted carbon emissions in 2035 and 2060 based on a land-use scenario. In addition to land-use changes, population and economic growths are also strongly associated with carbon emissions. Please describe demographic and economic assumptions used in the carbon emissions prediction.

Response: We apologize for not clearly presenting our scenarios description in the section 4.2. We have added this information in lines 624-627. We also added a table to exhibit the detail assumption for the scenario analysis.

We are grateful for your comments, and hope that our responses and the resulting changes in the manuscript will be satisfactory. However, we will be happy to work with you to resolve any remaining issues.

Reviewer 4 Report

Introduction - solid and good 2.1. Model framework. Figure 1, source not given, sources for other drawings also not given (?). Why did the authors skip of years 2015-2018? Reasons must be given. In the text, the authors indicate Table S7 and Tables S1 to S6, but there are no such tables in the manuscript (rows 159-162) and the following tables in the text ... "Where CEi, CEr, and CEl represents the carbon emission of energy consumption, respiration by the city's population and livestock breeding, and agriculture (carried by cultivated land)" - indicate which variable (precisely) which means (rows 183-184 ) "Based on the integrated input flow gradients shown in Figure S5" Figure S5 (row 498) is not in the text 

The conclusions can be extended.

Author Response

Review #4:

  1. 2.1. Model framework. Figure 1, source not given, sources for other drawings also not given (?).

Response: The sources of all drawings have been added in manuscript in lines 313-315.

  1. Why did the authors skip of years 2015-2018? Reasons must be given.

Response: The sentence has been revised as “We obtained data on land uses throughout Beijing from 1990 to 2018, at a 5-year interval for 1990–2015 and a 3-year interval for 2015–2018 at a spatial resolution of 30 m detailed in section 2.5 and summarized the areas during each of the years in this period (Table S7).” in lines 173-174.

  1. In the text, the authors indicate Table S7 and Tables S1 to S6, but there are no such tables in the manuscript (rows 159-162) and the following tables in the text ...

Response: We have provided all Tables in the revised Supporting Information file.

  1. "Where CEi, CEr, and CEl represents the carbon emission of energy consumption, respiration by the city's population and livestock breeding, and agriculture (carried by cultivated land)" - indicate which variable (precisely) which means (rows 183-184)

Response: The sentence has been revised as “where CEi represents the carbon emission of energy consumption, CEr, is the carbon emission of respiration by the city’s population and livestock breeding, and CEl represents the carbon emission from agriculture (carried by cultivated land).” in lines 199-201.

  1. "Based on the integrated input flow gradients shown in Figure S5" Figure S5 (row 498) is not in the text 

Response: We have provided Figure S5 in the revised Supporting Information file.

  1. The conclusions can be extended.

Response: The conclusion part has been extended as suggested. In addition, we added a paragraph to describe the limitations of our estimation in lines 726-738.

We are grateful for your comments, and hope that our responses and the resulting changes in the manuscript will be satisfactory. However, we will be happy to work with you to resolve any remaining issues.

Round 2

Reviewer 2 Report

All my comments are answered and addressed in a satisfying way.